# A Toolkit for Reliable Benchmarking and Research in Multi-Objective Reinforcement Learning

**Florian Felten**[*,1,7]   **Lucas N. Alegre**[*,2,3,7]   **Ann Nowé**[3]   **Ana L. C. Bazzan**[2]
**El-Ghazali Talbi**[1,4]   **Grégoire Danoy**[1,5]   **Bruno C. da Silva**[6]

[1]SnT, University of Luxembourg    [2]Institute of Informatics, Federal University of Rio Grande do Sul
[3]Artificial Intelligence Lab, Vrije Universiteit Brussel    [4]CNRS/CRIStAL, University of Lille
[5]FSTM/DCS, University of Luxembourg    [6]University of Massachusetts    [7]Farama Foundation
{florian.felten,gregoire.danoy}@uni.lu   {lnalegre,bazzan}@inf.ufrgs.br
ann.nowe@vub.be   el-ghazali.talbi@univ-lille.fr   bsilva@cs.umass.edu

## Abstract

Multi-objective reinforcement learning algorithms (MORL) extend standard reinforcement learning (RL) to scenarios where agents must optimize multiple—potentially conflicting—objectives, each represented by a distinct reward function. To facilitate and accelerate research and benchmarking in multi-objective RL problems, we introduce a comprehensive collection of software libraries that includes: *(i)* MO-Gymnasium, an easy-to-use and flexible API enabling the rapid construction of novel MORL environments. It also includes more than 20 environments under this API. This allows researchers to effortlessly evaluate any algorithms on any existing domains; *(ii)* MORL-Baselines, a collection of reliable and efficient implementations of state-of-the-art MORL algorithms, designed to provide a solid foundation for advancing research. Notably, all algorithms are inherently compatible with MO-Gymnasium; and *(iii)* a thorough and robust set of benchmark results and comparisons of MORL-Baselines algorithms, tested across various challenging MO-Gymnasium environments. These benchmarks were constructed to serve as guidelines for the research community, underscoring the properties, advantages, and limitations of each particular state-of-the-art method.[2]

## 1   Introduction

Research in reinforcement learning (RL) algorithms (Sutton and Barto, 2018) has gained significant attention in recent years, in great part due to its remarkable success in a range of challenging problems (Mnih et al., 2015b; Silver et al., 2017; Bellemare et al., 2020). This led to a substantial increase in the number of papers published in the field every year. The rapid growth of RL research, however, has not necessarily been accompanied by the design of well-thought and reliable tools allowing for appropriate evaluation practices, resulting in a reproducibility crisis within the field. This resulted, e.g., in researchers often questioning the validity and reproducibility of results presented in influential papers (Agarwal et al., 2021; Patterson et al., 2023).

There are several reasons why reproducibility is often a challenge in RL. A key factor is the significant amount of time required to train RL agents, which makes it difficult for researchers to gather sufficient data to perform rigorous statistical analyses of empirical results. As a consequence, some authors may claim superior performance over state-of-the-art techniques without adequate evidence.

---

[*]Authors contributed equally to this work.

[2]MO-Gymnasium and MORL-Baselines are available at https://github.com/Farama-Foundation/mo-gymnasium and https://github.com/LucasAlegre/morl-baselines, respectively. The benchmark results are available on openrlbenchmark: https://wandb.ai/openrlbenchmark/MORL-Baselines.

37th Conference on Neural Information Processing Systems (NeurIPS 2023) Track on Datasets and Benchmarks.

Additionally, papers often do not provide sufficient information—such as hyperparameter values and implementation optimizations—to allow for the reliable reproducibility of their results. Finally, the use of environment implementations that lack standardization also contributes to this challenge. To address such issues, several libraries and experimentation protocols have been proposed in the RL ecosystem (Brockman et al., 2016; Agarwal et al., 2021; Raffin et al., 2021; Huang et al., 2022b).

As RL garners increasing interest, its subfield, Multi-Objective Reinforcement Learning (MORL), is simultaneously attracting notable attention within the RL research community (Hayes et al., 2022). MORL algorithms tackle problems where an agent has to optimize multiple—possibly conflicting—objectives. Each of these objectives is represented via a distinct reward function. In this setting, agents typically aim to find optimal decision-making policies defined with respect to different compromises or trade-offs between the objectives. Examples of MORL problems include the widely-used set of Mujoco (Todorov et al., 2012) tasks, which model compromises between an agent's objectives. As an example, the reward function of the *half-cheetah* agent is a weighted combination of velocity- and energy-related terms, where the weights are pre-determined and kept constant. However, allowing for such weights (i.e., for the relative importance of objectives) to change may induce significantly different optimal behaviors (Xu et al., 2020). While the goal of standard RL algorithms is to learn a single policy, specialized in optimizing a single reward function, MORL algorithms typically search for a *set* of policies such that given *any* compromises between objectives, a corresponding optimal (or near-optimal) policy is in the set. Designing and properly evaluating MORL algorithms shares the difficulties encountered when designing and empirically testing RL techniques—with additional challenges particular to multi-objective settings. Yet, to the best of our knowledge, there are currently no public standard libraries providing reliable implementations of widely-used MORL domains and state-of-the-art MORL algorithms, designed particularly to facilitate research in the field.

In this paper, we introduce a comprehensive suite of benchmark MORL environments and reliable implementations of widely-used MORL algorithms, designed to facilitate the construction of reproducible empirical performance evaluation of existing and novel multi-objective techniques. First, we introduce MO-Gymnasium (Section 4), an easy-to-use and flexible API enabling the rapid construction of novel MORL environments. As of now, this API includes over 20 environments with diverse characteristics. This allows researchers to evaluate, with minimal effort, any algorithms compatible with our extendable API in any existing domains. Secondly, we introduce MORL-Baselines (Section 5), a collection of reliable and efficient implementations of state-of-the-art MORL algorithms designed to provide a solid foundation for advancing MORL research. Notably, all such algorithms are inherently compatible with MO-Gymnasium. Finally, in Section 6 we provide a thorough and robust set of benchmark results and comparisons of MORL-Baselines algorithms tested across various challenging MO-Gymnasium environments. These benchmarks were constructed to serve as guidelines for the research community, underscoring the properties, advantages, and limitations of each particular state-of-the-art method.

## 2 Related work

The challenge of reproducing experimental results in machine learning research is widely acknowledged. The significance of this issue has risen to such a degree that top-tier conferences like NeurIPS have implemented reproducibility programs aimed at improving the standards for conducting, communicating, and evaluating research in the field (Pineau et al., 2020). The RL field is directly affected by this issue, as highlighted by different authors. For example, Engstrom et al. (2020) show that optimizations at the implementation level of deep RL algorithms can be more impactful than experimenting with different algorithms. Similarly, Huang et al. (2022a) investigated and identified the existence of 37 particular code-level tricks necessary to achieve state-of-the-art results when deploying the PPO algorithm (Schulman et al., 2017a). Agarwal et al. (2021) point out the lack of statistically significant results in numerous papers published in the field, despite many of them claiming to introduce techniques that outperform the state-of-the-art. Finally, an in-depth discussion of issues pertaining to methodologies used to conduct empirical evaluation in RL is presented by Patterson et al. (2023). These issues include, among others, inadequate tuning of the hyperparameters of baselines algorithms when deployed on new environments, averaging performance metrics over a limited number of runs, absence of random seed control, and environment overfitting.

To address these concerns, several RL libraries have been developed to provide more reliable baselines. For example, Gymnasium (Towers et al., 2023) (formerly known as OpenAI Gym (Brockman et al.,

2016)) provides a standard API and a collection of reference environments for RL research and experimentation. Furthermore, various libraries have been published that contain well-tested, reliable, and continually maintained implementations of RL algorithms; e.g., Stable-Baselines 3 (Raffin et al., 2021) and cleanRL (Huang et al., 2022b). Finally, recent initiatives such as openrlbenchmark (Huang et al., 2023) facilitate the analysis of various learning metrics, and tackle the reproducibility challenge via experiment tracking software such as Weights and Biases, which makes it easier to, e.g., make hyperparameters of different baselines publicly available (Biewald, 2020). Other subfields of RL, such as multi-agent RL (MARL), have also benefited from standard baseline libraries. In MARL, PettingZoo (Terry et al., 2021) is often used to design novel environments, and EPyMARL (Papoudakis et al., 2021) is commonly used when designing new learning algorithms. Zhu et al. (2023) recently introduced D4MORL, a repository that includes datasets specifically designed to evaluate MORL algorithms in the *offline* setting. Although these subareas of RL have reaped the benefits from standardized libraries, to the best of our knowledge, there are currently no publicly available (and widely adopted) standard libraries providing reliable implementations of widely-used MORL domains and state-of-the-art MORL algorithms, designed to facilitate research in the field.

As discussed by Hayes et al. (2022); Cassimon et al. (2022), various benchmark problems have been proposed to evaluate MORL methods. However, these benchmarks have not yet been made available via standardized APIs or centralized repositories. Arguably, this has made the experimental reproducibility of MORL algorithms harder, time-consuming, and error-prone. MORL-Glue (Vamplew et al., 2017) represents an attempt to establish a centralized repository of MORL benchmarks. However, this library has not been widely adopted due to the fact that it is implemented in Java and targets tabular problems, whilst the community currently focuses on using Python and deep RL techniques. Our first contribution, MO-Gymnasium, addresses this issue by introducing a standard API and set of reference environments for MORL.

Although some published works on MORL make their code publicly available (Yang et al., 2019; Abels et al., 2019; Xu et al., 2020), these implementations are not often maintained by their authors. This makes reproducing empirical results a usually time-consuming and error-prone process. Our second contribution, MORL-Baselines, tackles this challenge. It includes a set of clear, well-structured, regularly maintained, and reliable implementations of MORL algorithms. Importantly, all such algorithms are compatible with MO-Gymnasium's API.

Finally, recall that the performance of newly proposed algorithms is generally compared with previously-published baselines. However, the methodology employed in conducting experiments is not always scientifically rigorous. To address this issue, we introduce a dataset of training results of MORL-Baselines algorithms when evaluated on all MO-Gymnasium environments. This dataset is available through openrlbenchmark and includes information about all hyperparameters used by each algorithm, in each experiment, thereby allowing researchers to compare new algorithms with such existing baselines without having to retrain models from scratch.

## 3 Multi-objective reinforcement learning

In MORL, the interaction of an agent with its environments is modeled via a *multi-objective Markov decision process* (MOMDP) (Roijers et al., 2013). MOMDPs differ from standard MDPs (Sutton and Barto, 2018) only in that a MOMDP's reward function is vector-valued. A MOMDP is defined as a tuple $M \triangleq (\mathcal{S}, \mathcal{A}, p, \mathbf{r}, \mu, \gamma)$, where $\mathcal{S}$ is a state space, $\mathcal{A}$ is an action space, $p(\cdot|s, a)$ is the distribution over next states given state $s$ and action $a$, $\mathbf{r} : \mathcal{S} \times \mathcal{A} \times \mathcal{S} \mapsto \mathbb{R}^m$ is a multi-objective reward function containing $m$ objectives, $\mu$ is an initial state distribution, and $\gamma \in [0, 1)$ is a discounting factor. A policy $\pi : \mathcal{S} \mapsto \mathcal{A}$ is a function mapping states to actions. Let $S_t$, $A_t$, and $\mathbf{R}_t = \mathbf{r}(S_t, A_t, S_{t+1})$ denote the random variables corresponding to state, action, and vector reward, respectively, at time step $t$. The *multi-objective value function* of a policy $\pi$ in state $s$ is defined as

$$\mathbf{v}^\pi(s) \triangleq \mathbb{E}_\pi \left[ \sum_{i=0}^{\infty} \gamma^i \mathbf{R}_{t+i} \mid S_t = s \right], \tag{1}$$

where $\mathbb{E}_\pi[\cdot]$ denotes expectation with respect to trajectories induced by the policy $\pi$. We denote $\mathbf{v}^\pi \triangleq \mathbb{E}_{S_0 \sim \mu}[\mathbf{v}^\pi(S_0)]$ as the *value vector* of policy $\pi$. Notice that $\mathbf{v}^\pi$ is an $m$-dimensional vector whose $i$-th component is the expected return of $\pi$ under the $i$-th objective.

In contrast to single-objective RL, comparing the values of two different policies is not straightforward in MORL. For instance, a policy that achieves higher expected return with respect to one objective may, as a result, have lower performance with respect to other objectives. Hence, in MORL, policies are typically evaluated and compared in terms of a *user utility function* (or scalarization function), $u : \mathbb{R}^m \mapsto \mathbb{R}$, which is a mapping from the multi-objective value of policy $\pi$, $\mathbf{v}^\pi$, to a scalar.

Let $u$ be a monotonically increasing utility function; i.e., a function such that the utility increases if the value of one objective can be improved without decreasing the value of other objectives. Then, the value vector $\mathbf{v}^\pi$ of an optimal policy with respect to $u$ is in a *Pareto frontier* (PF). The PF of a MOMDP is a set of nondominated multi-objective value vectors:

$$\mathcal{F} \triangleq \{\mathbf{v}^\pi \mid \nexists \pi' \text{s.t. } \mathbf{v}^{\pi'} \succ_p \mathbf{v}^\pi\}, \text{ where } \mathbf{v}^\pi \succ_p \mathbf{v}^{\pi'} \iff (\forall i : v_i^\pi \geq v_i^{\pi'}) \wedge (\exists i : v_i^\pi > v_i^{\pi'}). \quad (2)$$

In particular, *linear utility functions* linearly combine the value of a policy under each of the $m$ objectives using a set of weights $\mathbf{w} \in \mathbb{R}^m$: $u(\mathbf{v}^\pi, \mathbf{w}) \triangleq \mathbf{v}^\pi \cdot \mathbf{w}$, where each element of $\mathbf{w} \in \mathbb{R}^m$ specifies the relative importance of each objective. The space of weight vectors, $\mathcal{W}$, is an $m$-dimensional simplex so that $\sum_i w_i = 1, w_i \geq 0, i = 1, ..., m$. Under such a utility scheme, the PF corresponds to a *convex coverage set* (CCS), as shown by Roijers et al. (2013).

MORL algorithms can be classified in terms of whether they are *single-policy* (i.e., they learn a single policy that optimizes one particular utility $u$), or *multi-policy* (i.e., they learn a set of policies with the goal of approximating the PF). When non-linear utility functions are considered, the optimization procedure of a MORL problem can also be defined with respect to the *expected scalarized return* (ESR), $\mathbb{E}[u(\sum_{t=0}^\infty \gamma^t \mathbf{R}_t)]$, or the *scalarized expected return* (SER), $u(\mathbb{E}[\sum_{t=0}^\infty \gamma^t \mathbf{R}_t])$. For a thorough review of MORL concepts and definitions, we refer the reader to Hayes et al. (2022).

## 4  MO-Gymnasium

In this section, we introduce our first contribution: MO-Gymnasium.[3] MO-Gymnasium is an easy-to-use and flexible API enabling the rapid construction of novel MORL environments, and a collection of regularly maintained and thoroughly tested environment implementations.

```
import mo_gymnasium as mo_gym
env = mo_gym.make('minecart-v0')
obs, info = env.reset()
for _ in range(1000):
    action = policy(obs)
    obs, rews, term, trunc, info = env.step(action)
```

Figure 1: An example of how MO-Gymnasium can be used.

Unlike classical ML settings, which rely on fixed datasets, RL problems typically do not; this makes replication of experimental results challenging. Indeed, even small differences in the definition of environment can have a significant impact on the performance of algorithms.

To address this issue and facilitate research when it comes to the standard RL settings, Gymnasium (Towers et al., 2023) (formerly Gym (Brockman et al., 2016)) introduces an API and collection of versioned environments. With millions of downloads, this is currently the *de facto* standard library in RL, enabling researchers to easily test their algorithms on a variety of problems. Despite its widespread use, Gymnasium is limited to modeling single-objective MDPs. It has since been expanded in various ways; e.g., PettingZoo extends it to MARL settings and D4MORL to offline MORL. To the best of our knowledge, there are currently no publicly available and widely adopted libraries providing reliable implementations of MORL domains and state-of-the-art MORL algorithms, designed to facilitate research in the field.

We address this limitation by introducing MO-Gymnasium—previously known as MO-Gym (Alegre et al., 2022b). MO-Gymnasium's API is designed to be as similar as possible to Gymnasium's API. This allows it to inherit many of the features in Gymnasium, such as *wrappers*—features that allow individual properties of a domain to be modified—while extending the original API only where and when necessary. This makes MO-Gymnasium automatically backward compatible with a wide range of MORL benchmark domains. The key difference between these two frameworks is that, in MO-Gymnasium, rewards returned after the execution of an action (i.e., after a call of the *step* method) are vectors rather than scalars (see Figure 1). MO-Gymnasium is available on PyPI and can be installed via `pip install mo-gymnasium`. Importantly, we highlight that MO-Gymnasium is

---

[3]The documentation of MO-Gymnasium is available at https://mo-gymnasium.farama.org.

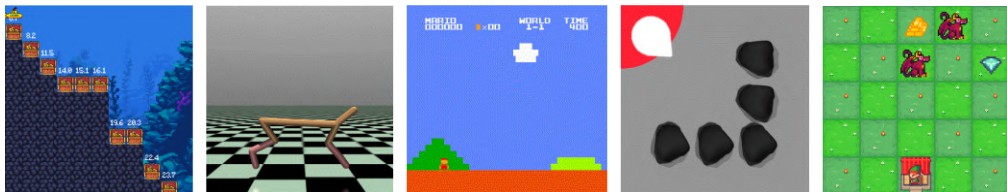

Figure 2: A few of the environments available in MO-Gymnasium. From left to right: *deep-sea-treasure*, *mo-halfcheetah*, *mo-supermario*, *minecart*, and *resource-gathering*.

an official part of the projects maintained by the Farama Foundation, and is considered a mature and well-supported library by the research community.

**Environments.** Currently, MO-Gymnasium includes over 20 environments commonly used in the MORL literature—including environments with discrete and continuous state and action spaces—such as *deep-sea-treasure* (Vamplew et al., 2011), *four-room* (Alegre et al., 2022a), *mo-supermario* (Yang et al., 2019), *minecart* (Abels et al., 2019), and *mo-halfcheetah* (Xu et al., 2020). In Figure 2 we depict a few of the currently available environments. See Appendix B for a detailed description of each environment. This large collection of environments allows designers to thoroughly assess the performance of novel algorithms in different scenarios. In environments in which the true PF is known, it can be accessed via the `pareto_front()` method available in the MO-Gymnasium's API. Additionally, for reproducibility purposes, each environment is labeled with a version number; e.g., *"-v0"*. Each time an environment is modified in a way that may affect algorithms' performances, the domain's version number is incremented.

**Wrappers.** MO-Gymnasium introduces MORL-specific wrappers such as *MONormalizeReward*, which normalizes a given component of the reward vector; and *LinearReward*, a wrapper that linearly scalarizes the reward function of a MOMDP environment, transforming it into a standard MDP. The latter feature makes MO-Gymnasium *directly compatible* with widely-used RL libraries compatible with Gymnasium, such as Stable-Baselines 3 (Raffin et al., 2021) and cleanRL (Huang et al., 2022b).

## 5 MORL-Baselines

In this section, we introduce MORL-Baselines[4], a collection of reliable and efficient implementations of state-of-the-art MORL algorithms, designed to provide a solid foundation for advancing research in MORL. To the best of our knowledge, this stands as the first open-source repository encompassing a multitude of MORL algorithms. Notably, all such algorithms are inherently compatible with MO-Gymnasium's API.

It is well known that the performance of learning algorithms is closely tied to their specific implementations (Engstrom et al., 2020). Unfortunately, details such as implementation-specific optimizations are seldom discussed in research papers, which makes reproducibility and comparisons challenging. To address this issue, some authors provide access to their codebases. While this is a positive step, codebases are often not regularly maintained and may become outdated, rendering replication difficult. Thus, libraries like Stable-Baselines 3 (Raffin et al., 2021) and cleanRL (Huang et al., 2022b) have been designed with the goal of regularly maintaining state-of-the-art algorithms. These codebases are well documented, thoroughly tested, and offer top-notch performance on the algorithms they implement. This allows researchers to start coding novel methods by extending existing implementations rather than starting from scratch. These libraries also provide useful tools for research purposes, such as efficient replay buffer implementations, performance reports, and evaluation methods.

MORL-Baselines includes more than 10 state-of-the-art MORL algorithms, all of which are compatible with MO-Gymnasium. Up to this point, no similar libraries were available. MORL-Baselines offers a range of features to aid researchers in designing new algorithms, such as methods to compute and analyze Pareto fronts, perform evaluation w.r.t. various metrics, replay buffers, and experiment tracking tools. Below, we provide a list of the algorithms currently available via MORL-Baselines, along with a description of the settings in which they are applicable.

---

[4]MORL-Baselines documentation is available at: `https://lucasalegre.github.io/morl-baselines`.

Table 1: Algorithms currently implemented in MORL-Baselines. (*) PCN and PQL are designed to tackle deterministic environments. (**) OLS is an algorithm-agnostic method for generating reward weights, or preferences; it does not assume any particular type of observation or action spaces.

| Algorithm | Single or multi-policy | Utitlity function | Observation space | Action space |
|---|---|---|---|---|
| MOQL (Van Moffaert et al., 2013a) | Single | Linear | Disc. | Disc. |
| EUPG (Roijers et al., 2018a) | Single | Non-linear, ESR | Disc. | Disc. |
| MPMOQL (Van Moffaert et al., 2013a) | Multi | Linear | Disc. | Disc. |
| PQL (Van Moffaert and Nowé, 2014) | Multi | Non-linear, SER (*) | Disc. | Disc. |
| OLS (Roijers, 2016) | Multi | Linear | / (**) | / (**) |
| Envelope (Yang et al., 2019) | Multi | Linear | Cont. | Disc. |
| PGMORL (Xu et al., 2020) | Multi | Linear | Cont. | Cont. |
| PCN (Reymond et al., 2022) | Multi | Non-linear, ESR/SER (*) | Cont. | Disc. |
| GPI-LS & GPI-PD (Alegre et al., 2023) | Multi | Linear | Cont. | Any |
| CAPQL (Lu et al., 2023) | Multi | Linear | Cont. | Cont. |

## 5.1 Implemented algorithms

Table 1 lists the algorithms currently supported by MORL-Baselines and the MORL settings they tackle. Algorithms employing neural networks as function approximators were implemented using PyTorch (Paszke et al., 2019), and tabular algorithms rely on NumPy (Harris et al., 2020). Algorithms in Table 1 are described according to whether they produce a single policy (based on user-provided utility functions) or multiple policies (i.e., to approximate a CCS or a PF). Additionally, notice that some algorithms optimize w.r.t. the ESR while others optimize w.r.t. SER (Section 3). Finally, notice that MORL-Baselines' algorithms may support different observation and action spaces (e.g., images).

**Tabular algorithms.** Multi-objective Q-learning (MOQL) (Van Moffaert et al., 2013b) is an extension of the classic tabular Q-learning algorithm (Watkins, 1989) that learns and stores the Q-values of each objective separately. A scalarization function is then used to convert these Q-values into a scalar quantity, allowing agents to select an action. Multi-policy MOQL, which consists of running MOQL multiple times with different preferences, can be instantiated using different methods that select which preference weight vector will be optimized next by a corresponding specialized policy. Currently, MORL-Baselines supports randomly generated weight vectors, Optimistic Linear Support (OLS) (Roijers, 2016), and Generalized Policy Improvement Linear Support (GPI-LS) (Alegre et al., 2023). Pareto Q-learning (PQL) (Van Moffaert and Nowé, 2014) aims at simultaneously learning all policies in the Pareto front by storing sets of non-dominated Q-values. These sets are then converted into scalars (using metrics similar to the ones discussed in the next section) that guide the selection of actions during the learning phase. This algorithm is only compatible with deterministic environments.

**Deep MORL algorithms.** The Expected Utility Policy Gradient (EUPG) algorithm (Roijers et al., 2018b) introduced a policy gradient update capable of taking into account both the return achieved up to the current moment, as well as future returns, in order to determine expected utilities in an ESR setting. However, although it employs a neural network as the policy, it has only been

evaluated in discrete settings. The Envelope algorithm (Yang et al., 2019) uses a single neural network conditioned on a weight vector (Abels et al., 2019) to approximate the CCS. Prediction-Guided MORL (PGMORL) (Xu et al., 2020) is an evolutionary algorithm that maintains a population of policies learned using PPO (Schulman et al., 2017b). This algorithm focuses on predicting, at each iteration, the most promising weight vectors and policies to select for further training in order to more effectively enhance the PF. Pareto Conditioned Networks (PCN) (Reymond et al., 2022) employs a neural network conditioned on a given desired return per objective. This network is trained via supervised learning to predict which actions produce the desired return in deterministic environments. GPI-LS (Alegre et al., 2023) employs GPI (Barreto et al., 2017) to combine policies in its learned CCS and prioritize the weight vectors on which agents should train at each moment. GPI-Prioritized Dyna (GPI-PD) is a model-based extension of GPI-LS that uses a learned model of the environment and GPI to prioritize experiences in the replay buffer. Concave-Augmented Pareto Q-learning (CAPQL) (Lu et al., 2023) employs a multi-objective extension of SAC (Haarnoja et al., 2018) by conditioning the actor and critic networks on the rewards weight vector.

## 5.2 Evaluation metrics

Recall that in single objective RL settings, policies are evaluated in terms of their corresponding expected returns. In MORL settings, by contrast, they are typically evaluated using multi-objective metrics computed based on Pareto fronts. While these metrics are widely adopted by the multi-objective optimization community, the MORL community has yet to establish a consensus on which metrics should be preferred in each particular problem or setting. For this reason, our framework supports all commonly-used MORL metrics—see below. These metrics can be split into two groups: utility-based metrics, which assume particular properties of the utility function (e.g., linearity), and axiomatic metrics, which do not make assumptions but may produce less informative performance information to users. For a thorough discussion of these metrics, see Hayes et al. (2022).

**Expected utility (↑).** If the utility function $u$ is linear, it is possible to express the expected utility over a distribution of rewards weights, $\mathcal{W}$, via the *expected utility* (EU) metric (Zintgraf et al., 2015). Let $\Pi$ be a set of policies and $\tilde{\mathcal{F}} = \{\mathbf{v}^\pi | \pi \in \Pi\}$ be its corresponding approximate PF. Then, the EU metric is defined as:

$$\text{EU}(\tilde{\mathcal{F}}) = \mathbb{E}_{\mathbf{w} \sim \mathcal{W}} \left[ \max_{\mathbf{v}^\pi \in \tilde{\mathcal{F}}} \mathbf{v}^\pi \cdot \mathbf{w} \right].$$

**Maximum utility loss (↓).** Zintgraf et al. (2015) introduced this metric to quantify the maximum utility loss that results from using an approximate PF, $\tilde{\mathcal{F}}$, rather than a reference PF, $\mathcal{Z}$[5]. It is defined as follows:

$$\text{MUL}(\tilde{\mathcal{F}}, \mathcal{Z}) = \max_{\mathbf{w} \in \mathcal{W}} (\max_{\mathbf{v}^* \in \mathcal{Z}} \mathbf{v}^{\pi^*} \cdot \mathbf{w} - \max_{\mathbf{v}^\pi \in \tilde{\mathcal{F}}} \mathbf{v}^\pi \cdot \mathbf{w}).$$

**Inverted generational distance (↓).** This metric characterizes the convergence rate of an approximate PF, $\tilde{\mathcal{F}}$, towards a reference PF, $\mathcal{Z}$ (Coello Coello and Reyes Sierra, 2004). If the reference front is unknown, it is usually defined/constructed by aggregating the best value vectors observed after several executions of the underlying RL algorithm. The inverted generation distance (IGD) is computed as:

$$\text{IGD}(\tilde{\mathcal{F}}, \mathcal{Z}) = \frac{1}{|\mathcal{Z}|} \sqrt{\sum_{\mathbf{v}^* \in \mathcal{Z}} \min_{\mathbf{v}^\pi \in \tilde{\mathcal{F}}} \|\mathbf{v}^* - \mathbf{v}^\pi\|^2}.$$

**Sparsity (↓).** This metric characterizes the diversity of the policies in a given PF. A diverse set of policies allows users to choose from qualitatively different behaviors based on the trade-offs they may wish to optimize (Xu et al., 2020). The sparsity of an approximate PF, $\tilde{\mathcal{F}}$, is given by:

$$\text{S}(\tilde{\mathcal{F}}) = \frac{1}{|\tilde{\mathcal{F}}| - 1} \sum_{j=1}^{m} \sum_{i=1}^{|\tilde{\mathcal{F}}|-1} (\mathcal{L}_j(i) - \mathcal{L}_j(i+1))^2,$$

where $\mathcal{L}_j$ is the sorted list of the values of the $j$-th objective considering all policies in $\tilde{\mathcal{F}}$, and $\mathcal{L}_j(i)$ is the $i$-th value in $\mathcal{L}_j$.

---

[5]If the true PF of the MOMDP, $\mathcal{F}$, is known, then typically $\mathcal{Z} = \mathcal{F}$.

**Hypervolume (↑).** This is a hybrid metric quantifying both a PF's rate of convergence and the diversity of the policies in it. Given an approximate PF, $\tilde{\mathcal{F}}$, and a reference point, $\mathbf{v}_{\text{ref}}$, the hypervolume metric (Zitzler, 1999) is defined as:

$$\text{HV}(\tilde{\mathcal{F}}, \mathbf{v}_{\text{ref}}) = \bigcup_{\mathbf{v}^\pi \in \tilde{\mathcal{F}}} \text{volume}(\mathbf{v}_{\text{ref}}, \mathbf{v}^\pi),$$

where $\text{volume}(\mathbf{v}_{\text{ref}}, \mathbf{v}^\pi)$ is the volume of the hypercube spanned by the reference vector, $\mathbf{v}_{\text{ref}}$, and the vector, $\mathbf{v}^\pi$. The reference point used in the hypervolume computation is typically an estimate of the worst-possible return per objective. For instance, one can use as reference point the vector $\frac{\mathbf{r}_{\min}}{1-\gamma}$, where $\mathbf{r}_{\min}$ is a vector with the minimum value for each objective in any given possible state.

### 5.3 Additional features

In addition to providing implementations of the algorithms and evaluation metrics discussed above, MORL-Baselines also provides a range of tools to assist researchers in designing new algorithms. These resources include a data structure for storing Pareto fronts and Pareto filtering functions, methods for policy evaluation with automated metric and PF reporting via Weights and Biases (Biewald, 2020), scalarization functions, various neural network architectures, as well as methods for generating videos of policies trajectories (e.g., via GIF files). For a comprehensive discussion of these functionalities, we refer the reader to Appendix C.2.

## 6 Benchmark dataset

Our final contribution is a comprehensive collection of benchmark results evaluating MORL-Baselines algorithms on a diverse set of MO-Gymnasium environments.[6] These results are compiled in a dataset hosted in and integrated into the openrlbenchmark framework (Huang et al., 2023). Access to such a dataset enables users performing new experiments to combine, compare, and aggregate the corresponding new results with the ones in the dataset, visualizing them in real-time using Weights and Biases (W&B) dashboards. They also allow users to manipulate raw data based on subsequent analyses, compare performance metrics of new algorithms w.r.t. state-of-the-art techniques without having to retrain the latter, and store the associated hyperparameters and command lines for each run. For instance, users can access the approximate Pareto fronts identified by each algorithm in our benchmark via the W&B dashboard, under the panel named 'eval/front'.

### 6.1 Proof-of-concept experiments using MORL-Baselines and MO-Gymnasium

In this section, we use MORL-Baselines and MO-Gymnasium to perform various proof-of-concept empirical experiments and discuss their corresponding results. Our goal is to highlight the type of experiments, comparisons, and analyses—of different algorithms on various domains, and under different performance metrics—that are possible using our framework. Given the potentially high cost of training algorithms, we performed 10 runs of each algorithm on each environment. Notice that while experimental results with tighter confidence intervals may be conducted (by performing more runs), our goal here is solely to highlight the capabilities of our libraries—not to fully evaluate all existing algorithms. The hyperparameters used on each run can be found in the W&B dashboards.[7] Examples of the W&B user interface through which our benchmark dataset can be analyzed are shown in the appendices.[8] Notice that some algorithms were trained over more timesteps than others since they may be fast but not as sample efficient; e.g., PGMORL relies on PPO (Schulman et al., 2017a), which is computationally faster but less sample efficient than GPI-LS, which relies on TD3 (Fujimoto et al., 2018a).

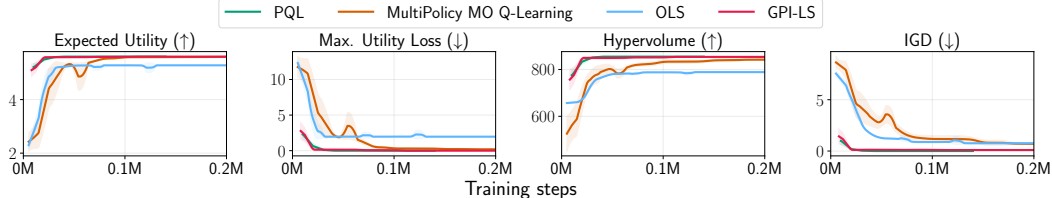

Figure 3: Performance of tabular MORL algorithms on *deep-sea-treasure-v0* w.r.t. training samples.

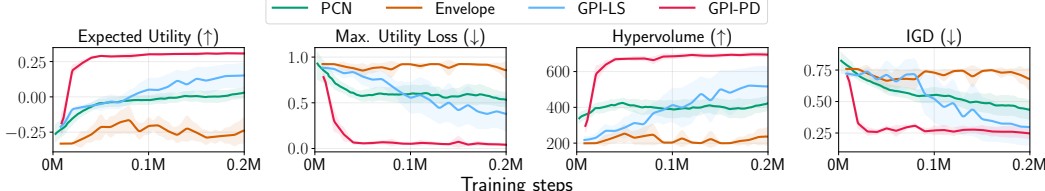

Figure 4: Performance of MORL algorithms on the *minecart-v0* domain w.r.t. training samples.

## 6.2 Results and discussion of the proof-of-concept experiments

Figures 3, 4, 5, and 6 present the performance of various MORL-Baselines algorithms when evaluated on a few representative MO-Gymnasium environments. They show, in particular, the mean and corresponding 95% confidence intervals with respect to the evaluation metrics introduced in Section 5.2. In the Appendix, we provide the complete set of experimental results and comparisons.

Figure 3 presents the performance of various tabular MORL algorithms available on MORL-Baselines, when evaluated in the classic *deep-sea-treasure* domain. Notice that the PQL and GPI-LS algorithms have similar performance and achieve near-zero maximum utility loss and Inverted Generational Distance. This indicates that they successfully identify the true Pareto front.

Figure 4 depicts the performance of various MORL algorithms that support discrete action spaces in *minecart-v0*. GPI-PD significantly outperforms all other baselines (when considering all performance metrics) in terms of sample efficiency and asymptotic performance. This empirical result supports recent observations by Alegre et al. (2022a) about the crucial role of efficient prioritization schemes when constructing PFs and selecting training experiences. Interestingly, notice that even though PCN was designed to tackle deterministic MOMDPs, it still outperforms the Envelope algorithm.

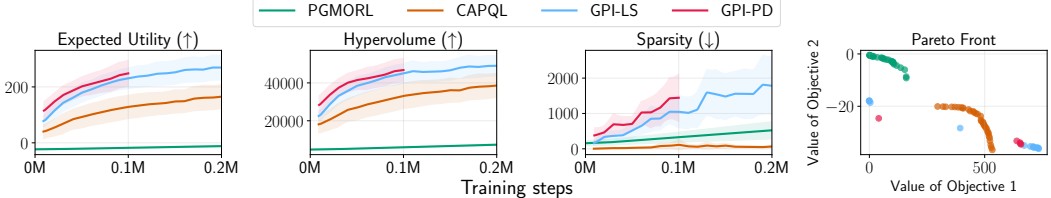

Figure 5: Peformance of MORL algorithms on the *mo-halfcheetah-v4* domain. Pareto fronts were constructed by identifying (across all runs) the front with the highest Expected Utility after a given training budget (5M steps for PGMORL, 100k for GPI-PD, 200k for the others).

Finally, Figures 5 and 6 present the performance of MORL algorithms that support continuous action spaces in the *mo-halfcheetah-v4* and *mo-hopper-2d-v4* domains, respectively. In *mo-halfcheetah*,

---

[6]These results can be viewed and analyzed at https://wandb.ai/openrlbenchmark/MORL-Baselines.

[7]We also include a description of the high-performance computers at the University of Luxembourg (Varrette et al., 2014) and Vrije Universiteit Brussel on which experiments were conducted. Training all algorithms on all environments, using various random seeds, required approximately 3 months of computation time.

[8]An overview of the command lines used to conduct each experiment can be found at https://github.com/LucasAlegre/morl-baselines/issues/43.

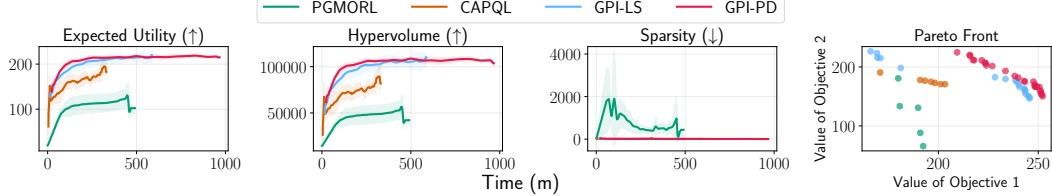

Figure 6: Performance of MORL algorithms on *mo-hopper-2d-v4* w.r.t. training time (in minutes).

GPI-based algorithms perform better w.r.t. the hypervolume and expected utility metrics. PGMORL and CAPQL, by contrast, perform better w.r.t. the Sparsity metric—they are able to identify denser Pareto fronts, as shown in the rightmost plot. Recall that sample efficiency often comes at the cost of execution time. To investigate this trade-off, we show, in Figure 6, performance results in the *mo-hopper* domain with respect to training time (in minutes). These experiments suggest that PGMORL performs well in terms of training time, in contrast to previous experiments where it had a qualitatively different behavior when analyzed with respect to sample complexity. In these experiments, both GPI-based algorithms still dominated all other baselines.

## 7 Conclusion

We introduced a comprehensive collection of software libraries for reliable benchmarking and research in MORL. MO-Gymnasium and MORL-Baselines are the first libraries providing standardized APIs and extensible sets of environments and state-of-the-art MORL algorithms. Our framework also includes a thorough set of benchmark results comparing a wide range of algorithms in various environments. These can be used as guidelines by the community, underscoring the properties and limitations of different techniques. By making this toolkit open source and extensible—and thus open to contributions from other researchers—we hope to provide a solid foundation for reproducible research in MORL. This work will allow researchers and end users to seamlessly deploy existing algorithms on various MORL domains, speeding up experiments, facilitating algorithm evaluation, and being more conducive to reproducible experimental results.

**Limitations and future work.** In the benchmarks we presented in this paper, we did not perform exhaustive hyperparameter tuning. Our main goal with these experiments was to validate our implementations and showcase the learning behavior of the algorithms in our framework with respect to different evaluation metrics. For this reason, such empirical results should be interpreted as proof-of-concept and may still be further improved with the aid of the MORL community. In future work, we will continue maintaining all algorithms in MORL-Baselines, and plan to augment the set of techniques available through our framework. We also plan to introduce new features for automating hyperparameter tuning. We do not anticipate any negative societal impacts of this work.

## Acknowledgments and disclosure of funding

We would like to thank Willem Röpke for his implementation of PQL, Denis Steckelmacher and Conor F. Hayes for the original implementation of EUPG, and Mathieu Reymond for the original implementation of PCN. Jordan K. Terry, Mark Towers, Manuel Goulão, and the broader Farama Foundation team for supporting MO-Gymnasium. Shengyi Huang, Antonin Raffin, and Quentin Gallouédec for their advice on experimental setup and openrlbenchmark integration. This work was funded by: Coordenação de Aperfeiçoamento de Pessoal de Nível Superior - Brazil (CAPES) - Finance Code 001; CNPq (Grants 140500/2021-9, 304932/2021-3); FAPESP/MCTI/CGI (Grant 2020/05165-1); the Fonds National de la Recherche Luxembourg (FNR), CORE program under the ADARS Project, ref. C20/IS/14762457; the Research Foundation Flanders (FWO) [G062819N]; the AI Research Program from the Flemish Government (Belgium); and the Francqui Foundation.

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

# Appendix

This appendix contains supplementary material about MO-Gymnasium and MORL-Baselines. In Section A, we provide general information such as the links to our libraries, licenses, and maintenance plan. In Section B, we describe in detail all environments available in MO-Gymnasium, and in Section C we present all features of MORL-Baselines. Finally, in Section D, we provide in more detail the experimental setting and results of our benchmarks.

## A  General information

### A.1  Links

The MO-Gymnasium documentation is accessible at: https://mo-gymnasium.farama.org and its code repository at: https://github.com/Farama-Foundation/MO-Gymnasium.

The MORL-Baselines documentation is available at: https://lucasalegre.github.io/morl-baselines and its code repository at: https://github.com/LucasAlegre/morl-baselines.

The openrlbenchmark code and early documentation are available at: https://github.com/openrlbenchmark/openrlbenchmark.

Our training metrics and results, hosted by Weights and Biases, are available at: https://wandb.ai/openrlbenchmark/MORL-Baselines.

### A.2  Licenses

MO-Gymnasium, MORL-Baselines, and openrlbenchmark are under the MIT License. We, the authors, bear all responsibility in case of violation of rights.

### A.3  Maintenance

MO-Gymnasium is maintained by the authors, with the support of the Farama Foundation team.[9] MORL-Baselines is currently maintained by the authors, with the support of the community. openrlbenchmark is still under development, yet a large community of researchers is already using it to upload RL training data.

## B  MO-Gymnasium

Below, we provide a description of each environment currently available in MO-Gymnasium. Figure 7 presents a visualization of each environment. A comprehensive list and documentation of the environments available through the MO-Gymnasium API, along with the definition of their observation and action spaces, can be found on the documentation website: https://mo-gymnasium.farama.org/environments/all-environments. We have plans to introduce more challenging environments in the future.

### B.1  *deep-sea-treasure-v0*

This domain, originally proposed by Vamplew et al. (2011), is a classic MORL problem in which the agent is a submarine that must collect a treasure while taking into account a time penalty. At each step, the agent perceives its current location as $x, y$ coordinates in a $11 \times 11$ grid world, and its actions correspond to movement in one of the four cardinal directions. Hence, in this domain, the state space is defined as $\mathcal{S} = \{0, \ldots, 10\}^2$, and the action space is $\mathcal{A} = \{\text{up}, \text{down}, \text{left}, \text{right}\}$. We use the same treasures' values as defined by Yang et al. (2019). A version of this domain with a concave Pareto frontier (Vamplew et al., 2011) is also available with identifier *deep-sea-treasure-concave-v0*. The

---

[9]https://farama.org/team.

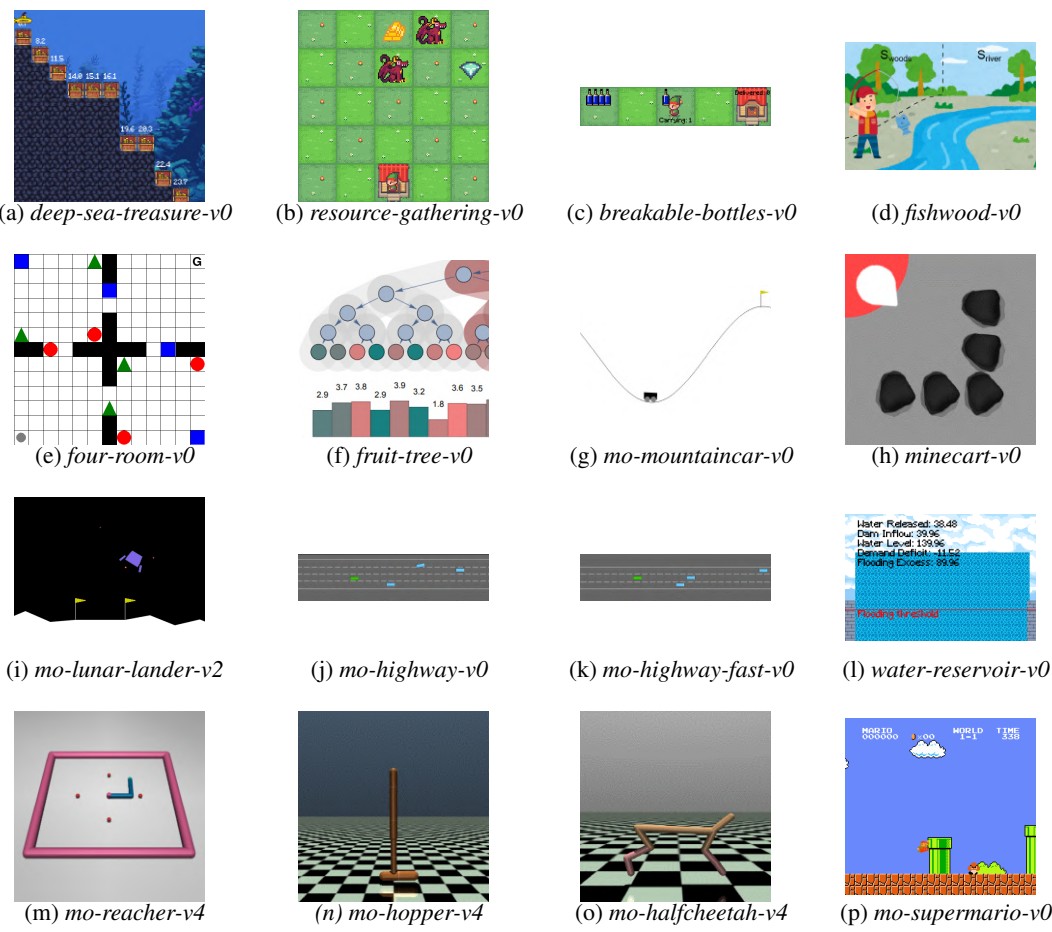

Figure 7: Visualization of the environments currently available in MO-Gymnasium.

multi-objective reward function of this domain, $\mathbf{r}(s, a, s') \in \mathbb{R}^2$, is defined as:

$$r_1(s, a, s') = \text{value of treasure in state } s',$$
$$r_2(s, a, s') = -1.$$

### B.2  *resource-gathering-v0*

In this domain, an agent must collect the gold or gem available in a $5 \times 5$ grid world while avoiding being killed by enemies (Barrett and Narayanan, 2008). The state space is defined as $\mathcal{S} = \{0, \ldots, 4\}^2 \times \{0, 1\}^2$ and denotes the agent's current position and two binary variables representing whether the agent carries gold or diamond. Its action space representing movements in the four cardinal directions is $\mathcal{A} = \{\text{up}, \text{down}, \text{left}, \text{right}\}$. There are two enemies that have a 10% chance of killing the agent if it is in the same cell. The multi-objective reward function of this domain, $\mathbf{r}(s, a, s') \in \mathbb{R}^3$, is defined as:

$$r_1(s, a, s') = -1 \text{ if killed by enemy in } s', \text{ else } 0,$$
$$r_2(s, a, s') = +1 \text{ if returned home with gold, else } 0,$$
$$r_3(s, a, s') = +1 \text{ if returned home with gem, else } 0.$$

### B.3  *breakable-bottles-v0*

This domain consists of a grid corridor with 5 cells. The agents must collect bottles from the source location and deliver them to the destination cell (Vamplew et al., 2021). Its action space is defined as $\mathcal{A} = \{\text{right}, \text{left}, \text{pick\_up\_bottle}\}$. Every time step, the agent has a 10% chance of dropping one of the

bottles it is currently carrying. The multi-objective reward function of this domain, $\mathbf{r}(s, a, s') \in \mathbb{R}^3$, is defined as:

$$r_1(s, a, s') = -1,$$
$$r_2(s, a, s') = +25 \times \#\text{bottles\_delivered},$$
$$r_3(s, a, s') = f(s') - f(s), \text{ where } f(s) = -1 \text{ if dropped any bottles in } s, \text{ else } 0.$$

### B.4 *fishwood-v0*

This is a simple MORL problem in which the agent controls a fisherman who can either fish or go collect wood (Roijers et al., 2018b). There are only two discrete states, and the agent can decide between fishing or collecting wood, *i.e.* $\mathcal{S} = \{\text{fishing}, \text{woods}\}$ and $\mathcal{A} = \{\text{go\_fish}, \text{go\_wood}\}$. The multi-objective reward function of this domain, $\mathbf{r}(s, a, s') \in \mathbb{R}^2$, is defined as:

$$r_1(s, a, s') = +1 \text{ if in the woods, with } 90\% \text{ probability, else } 0,$$
$$r_2(s, a, s') = +1 \text{ if in fishing, with } 10\% \text{ probability, else } 0.$$

### B.5 *four-room-v0*

This is a classic grid world environment in the literature of successor features (SFs) (Barreto et al., 2017). Importantly, Alegre et al. (2022a) showed that any problem in the SFs framework can be mapped to a MORL problem. In this domain, the agent can collect three different types of items (see Figure 7(e)). Its state space is defined by the $x, y$ coordinates of the agent as well as a binary vector stating which items have been collected, the actions are going into one of the four cardinal directions. Formally, $\mathcal{S} = \{0, \dots, 12\}^2 \times \{0, 1\}^{12}$, and $\mathcal{A} = \{\text{up}, \text{down}, \text{left}, \text{right}\}$. The multi-objective reward function of this domain, $\mathbf{r}(s, a, s') \in \mathbb{R}^3$, is defined as:

$$r_1(s, a, s') = +1 \text{ if collected a blue square (or is in goal location), else } 0,$$
$$r_2(s, a, s') = +1 \text{ if collected a green triangle (or is in goal location), else } 0,$$
$$r_3(s, a, s') = +1 \text{ if collected a red circle (or is in goal location), else } 0.$$

### B.6 *fruit-tree-v0*

This domain is modeled as a full binary tree of parameterized depth ($d \in \{5, 6, 7\}$) (Yang et al., 2019). The agent starts at the root node and can choose to move left or right until reaching a leaf node. The state space encodes the current position in the tree as a vector, which contains the current depth and the corresponding node: $\mathcal{S} = \{0, \dots, d\} \times \{0, \dots, 2^{d-1}\}$. The action space is defined as $\mathcal{A} = \{\text{left}, \text{right}\}$. Every leaf contains a fruit with a different value for the nutrients: protein, carbs, fats, vitamins, minerals, and water. The multi-objective reward function of this domain, $\mathbf{r}(s, a, s') \in \mathbb{R}^6$, is defined as:

$$r_i(s, a, s') = \text{value of nutrient } i \text{ in } s', \text{ for } i = 1 \dots 6.$$

### B.7 *mo-mountaincar-v0*

This domain, introduced by Vamplew et al. (2011), is a multi-objective version of the classic mountain car problem (Moore, 1990). The state space is defined by a vector stating the agent's current x-position and its velocity, $\mathcal{S} = [-1.2, 0.6] \times [-0.07, 0.07]$. Its actions consist in accelerating to the left, right, or doing nothing: $\mathcal{A} = \{\text{reverse}, \text{forward}, \text{none}\}$. Unlike the single-objective version, the agent is also penalized for applying the reverse and forward actions, which leads to different policies depending on the trade-offs between these penalties. The multi-objective reward function of this domain, $\mathbf{r}(s, a, s') \in \mathbb{R}^3$, is defined as:

$$r_1(s, a, s') = -1,$$
$$r_2(s, a, s') = -1 \text{ if } a = \text{reverse}, \text{ else } 0,$$
$$r_3(s, a, s') = -1 \text{ if } a = \text{forward}, \text{ else } 0.$$

A continuous action version of this domain is also available with identifier *mo-mountaincar-continuous-v0*. In this version, the action space is a speed vector $\mathcal{A} = [-1, 1]$, and the reward function contains two components: a time penalty and a fuel penalty.

$$r_1(s, a, s') = -1,$$
$$r_2(s, a, s') = -\|a\|^2.$$

## B.8  *minecart-v0*

This domain consists of a cart that must collect two different ores and return them to the base while minimizing fuel consumption (Abels et al., 2019). The agent perceives its environment through a 7-dimensional vector containing the agent $x, y$ position, the current speed of the cart, its orientation (sin and cos), and the percentage of occupied capacity in the cart by each ore: $\mathcal{S} = [-1, 1]^5 \times [0, 1]^2$. The agent has the choice between 6 actions: $\mathcal{A} = \{\text{mine, left, right, accelerate, brake, do nothing}\}$. Each mine in Figure 7(h) has a different distribution over two types of ores. Fuel is consumed at every time step, and extra fuel is consumed when the agent accelerates or selects the mine action. A version of this domain with deterministic values for the ores is also available as *minecart-deterministic-v0*. The multi-objective reward function of this domain, $\mathbf{r}(s, a, s') \in \mathbb{R}^3$, is defined as:

$$r_1(s, a, s') = \text{quantity of ore 1 collected if } s' \text{ is inside the base, else } 0,$$
$$r_2(s, a, s') = \text{quantity of ore 2 collected if } s' \text{ is inside the base, else } 0,$$
$$r_3(s, a, s') = -0.005 - 0.025\mathbb{1}\{a = \text{accelerate}\} - 0.05\mathbb{1}\{a = \text{mine}\}.$$

## B.9  *mo-lunar-lander-v2*

This domain is a multi-objective version of the lunar lander environment available in Gymnasium (Towers et al., 2023). The state is an 8-dimensional vector: the $x, y$ coordinates of the lander, its linear velocities in x & y, its angle, its angular velocity, and two booleans that represent whether each leg is in contact with the ground or not: $\mathcal{S} = [-1.5, 1.5]^2 \times [-5, 5] \times [-\pi, \pi] \times [-5, 5] \times \{0, 1\}^2$. The action space consists of four actions: $\mathcal{A} = \{\text{do nothing, fire left orientation engine, fire main engine, fire right engine}\}$. A continuous action version of this domain is also available with identifier *mo-lunar-lander-continuous-v2*. In this variant, the first coordinate of the action determines the throttle of the main engine, while the second coordinate specifies the throttle of the lateral boosters, $\mathcal{A} = [-1, 1]^2$. The multi-objective reward function of this domain, $\mathbf{r}(s, a, s') \in \mathbb{R}^4$, is defined as:[10]

$$r_1(s, a, s') = +100 \text{ if landed successfully}, \ -100 \text{ if crashed, else } 0,$$
$$r_2(s, a, s') = \text{shaping reward},$$
$$r_3(s, a, s') = \text{fuel cost of the main engine}$$
$$r_4(s, a, s') = \text{fuel cost of the side engines}.$$

## B.10  *mo-highway-v0*

This domain is a multi-objective version of the autonomous driving environment introduced by Leurent (2018). In this domain, the agent controls a vehicle on a multilane highway populated with other vehicles, it perceives its coordinates as well as coordinates from other vehicles. Formally, the state space is given by an array of size $V \times F$, where $V$ represents the number of vehicles in the environment and $F$ are the features describing each vehicle, *e.g.* velocity, position, angle. The agent's actions consist in changing lanes, accelerating or decelerating, or doing nothing, *i.e.* $\mathcal{A} = \{\text{lane left, idle, lane right, faster, slower}\}$. The multi-objective reward function of this domain, $\mathbf{r}(s, a, s') \in \mathbb{R}^3$, is defined as:

$$r_1(s, a, s') = \text{normalized forward speed of the vehicle},$$
$$r_2(s, a, s') = \text{bonus for driving in the rightmost lane},$$
$$r_3(s, a, s') = -1 \text{ if vehicle crashed, else } 0.$$

All three reward functions above are zeroed when the agent is off the road.

---

[10]See https://gymnasium.farama.org/environments/box2d/lunar_lander for details on the shaping reward.

## B.11 *mo-highway-fast-v0*

This is a more computationally efficient version of *mo-highway-v0* with reduced number of lanes and vehicles. The reward function, state, and action spaces are kept the same as the original environment.

## B.12 *water-reservoir-v0*

This domain simulates a water reservoir and is implemented as defined by Castelletti et al. (2012). The agent perceives a floating point number corresponding to the current amount of water in the reservoir: $\mathcal{S} \subset \mathbb{R}^+$. The agent executes a continuous action corresponding to the amount of water the dam will release in a day: $\mathcal{A} \subset \mathbb{R}^+$. The action space can also be specified to be normalized, in which case the action is expressed as a percentage of water to release: $\mathcal{A} = [0, 1]$. The multi-objective reward function of this domain, $\mathbf{r}(s, a, s') \in \mathbb{R}^4$, is defined as:

$$r_1(s, a, s') = \text{cost due to excess level w.r.t. a flooding threshold (upstream)},$$
$$r_2(s, a, s') = \text{deficit in the water supply w.r.t. the water demand},$$
$$r_3(s, a, s') = \text{deficit in hydroelectric supply wrt hydroelectric demand},$$
$$r_4(s, a, s') = \text{cost due to excess level w.r.t. a flooding threshold (downstream)}.$$

By default, only rewards $r_1$ and $r_2$ are used. However, the user has the option to instantiate the environment using all four reward functions.

## B.13 *mo-reacher-v4*

This domain is also a classic in the SFs literature (Barreto et al., 2017; Alegre et al., 2022a). It consists of a two-joint robot arm that must reach different target locations with the tip of its arm, implemented using the Mujoco robotics simulator (Todorov et al., 2012). The agent's state space $\mathcal{S} \subset \mathbb{R}^6$ consists of the sine and cosine of the angles of the central and elbow joints, as well as their angular velocities. The action space, originally continuous, is discretized using 3 bins per dimension corresponding to maximum positive torque (+1), negative torque (-1), and zero torque for each actuator. This results in a total of 9 possible actions: $\mathcal{A} = \{-1, 0, 1\}^2$. The multi-objective reward function $\mathbf{r}(s, a, s') \in \mathbb{R}^4$ is defined as:

$$r_i(s, a, s') = 1 - 4\Delta(\text{target}_i), \text{ for } i = 1...4, \tag{3}$$

where $\Delta(\text{target}_i)$ is the Euclidean distance between the tip of the robot's arm and the $i$-th target's location.

## B.14 *mo-hopper-v4*

This domain is a multi-objective version of the Gymnasium's *Hopper-v4* environment. The state space $\mathcal{S} \subset \mathbb{R}^{11}$ consists in velocities, angles, and position of each part of the robot. An action represents the torques applied at the hinge joints, $\mathcal{A} = [-1, 1]^3$. In this modified version, the agent must balance optimizing for its forward speed, jumping height, and energy cost. A version of this environment with a two-dimensional reward function is available with identifier *mo-hopper-2d-v4*. The multi-objective reward function $\mathbf{r}(s, a, s') \in \mathbb{R}^3$ is defined as:

$$r_1(s, a, s') = \text{velocity of the agent in the x-axis direction},$$
$$r_2(s, a, s') = \text{height of the agent over the z-axis},$$
$$r_3(s, a, s') = -||a||_2^2.$$

## B.15 *mo-halfcheetah-v4*

This domain is a multi-objective version of the Gymnasium's *HalfCheetah-v4* environment. The state space encodes the velocities, angles, and position of each part of the robot: $\mathcal{S} \subset \mathbb{R}^{17}$. Also, the actions are the torque to apply at each joint: $\mathcal{A} = [-1, 1]^6$. The multi-objective reward function $\mathbf{r}(s, a, s') \in \mathbb{R}^2$ is defined as:

$$r_1(s, a, s') = \text{forward speed of the agent},$$
$$r_2(s, a, s') = -||a||_2^2.$$

### B.16  *mo-supermario-v0*

This domain consists of the first level of the Super Mario Bros. video game. It is implemented using Gym Super Mario (Kauten, 2018). In this environment, the observations consist of RGB frames of the game, and the actions are the following combination of buttons: $\mathcal{A} = \{\text{NOOP}, \text{right}, \text{right} + A, \text{right} + B, \text{right} + A + B, A, \text{left}\}$. The multi-objective reward function $\mathbf{r}(s, a, s') \in \mathbb{R}^5$ is defined similarly as defined by Yang et al. (2019):

$$r_1(s, a, s') = \text{increase in the x-coordinate of Mario,}$$
$$r_2(s, a, s') = -\text{time elapsed executing action } a,$$
$$r_3(s, a, s') = -25 \text{ if Mario died, else } 0,$$
$$r_4(s, a, s') = 100 \times \text{\#coins collected,}$$
$$r_5(s, a, s') = \text{increase in score by killing an enemy.}$$

## C  MORL-Baselines

This section describes MORL-Baselines in more detail. MORL-Baselines is a comprehensive library that offers dependable, validated, well-documented, and efficient implementations of MORL algorithms. The library goes beyond supporting existing algorithm implementations by equipping researchers with valuable tools to streamline the process of designing future MORL algorithms. In the remainder of this section, we describe the algorithms, features, and provide a few examples of the user interface brought by this library.

### C.1  Algorithms

All the algorithms currently supported in MORL-Baselines are listed in Table 1 of the main paper. A few algorithms also have variants that the user can select by instantiating the agents with different parameters. In particular, our implementation of Multi-Policy MO Q-Learning (Van Moffaert et al., 2013a) can be derivated in multiple ways. For instance, the weights can be generated randomly, using OLS (Roijers, 2016), or using GPI-LS (Alegre et al., 2023). Moreover, using GPI (Barreto et al., 2017) to select actions can be enabled or not, as well as using model-based learning with GPI-PD (Alegre et al., 2023).

More details on the implementations are available in our documentation and GitHub repository. For example, see the documentation of PGMORL https://lucasalegre.github.io/morl-baselines/algos/multi_policy/pgmorl/.

### C.2  Additional features

Aside from the algorithms, MORL-Baselines provides a set of utilities allowing the user to construct novel MORL algorithms. These are listed below.

**Pareto frontier filters.** Our library provides a set of functions that permit to filter Pareto-dominated points out of a set of points. The signatures and documentation of these functions can be found in our documentation at https://lucasalegre.github.io/morl-baselines/features/pareto.

**Policies evaluation.** In order to build a Pareto front of policies, it is necessary to assess the values of the policies currently undergoing training in the environment. To standardize the evaluation process and enable fairer comparisons, we offer a collection of functions specifically designed for evaluating these policies. Furthermore, we have implemented a function that receives a Pareto front as input and records all metrics to Weights and Biases (W&B). This approach is employed across all multi-policy algorithms to ensure standardized metric logging. Additionally, we provide a method to initialize all components with a seed, including making PyTorch deterministic. The documentation related to these can be found at https://lucasalegre.github.io/morl-baselines/features/evaluations/.

**Performance metrics.** As outlined in the main paper, there are multiple metrics that can be used to evaluate a given Pareto front in order to compare different algorithms. We offer a collection of

metrics implementations for this purpose, including Hypervolume, Inverted Generational Distance, Maximum Utility Loss, Sparsity, and Expected Utility. These implementations accept a Pareto front as input and produce a scalar value representing the corresponding metric value. More details can be found at https://lucasalegre.github.io/morl-baselines/features/performance_indicators/.

**Scalarization functions.** Although most of the MORL algorithms rely on weighted sum scalarization, there are still cases where the user utility can be captured by non-linear schemes. To solve such an issue, we provide other scalarization schemes such as the Chebyshev scalarization function as used in Van Moffaert et al. (2013b). More details can be found at https://lucasalegre.github.io/morl-baselines/features/scalarization/.

**Reward weights utils.** There are many ways to generate weight vectors to scalarize vectorized rewards. These can, for instance, be randomly sampled or generated as equidistant points from the weight simplex, e.g., via the Riesz s-Energy method (Blank et al., 2021).[11] These helpers can be found at https://lucasalegre.github.io/morl-baselines/features/weights/.

**Experience replay buffers.** In RL algorithms, experience replay buffers play a crucial role as they store experienced tuples collected via interacting with the environment. These are usually used for updating the agent's policy and/or value functions via sampling. There are various approaches to designing these buffers. For instance, Abels et al. (2019) propose an experience replay buffer that maintains diverse samples to enable the learning of diverse policies. On the other hand, Alegre et al. (2023) rely on a prioritized experience replay buffer. We make these implementations available so that researchers can incorporate them into novel algorithms. More details can be found at https://lucasalegre.github.io/morl-baselines/features/buffers.

**Neural networks.** Neural networks are the main component used in modern deep MORL algorithms. There are many possible architectures and enhancements related to the usage of such function approximators, *e.g.* Convolutional Neural Networks are usually used on environments with image-based observations, and Polyak update is used when the algorithm relies on target networks. We provide a few helpers related to neural networks in MORL-Baselines; see https://lucasalegre.github.io/morl-baselines/features/networks/.

**Miscellaneous utils.** Finally, there are a couple of other methods that have been reused in MORL-Baselines. For example, we provide utilities to capture GIFs from the policy executions, or a linearly decaying value for epsilon-greedy exploration, as in Mnih et al. (2015a). All these utilities are listed at https://lucasalegre.github.io/morl-baselines/features/misc.

### C.3 User interface

This section gives some insight into the user interface provided by Weights and Biases to visualize and manipulate the training results.

Figure 8 shows an example of the real-time metrics obtained during the training. In this example we have a view of the multi-objective metrics described in the main paper, as well as a live view of the Pareto front found by various algorithms runs. Figure 9 illustrates the overview of a run. It provides information on the exact version of the library that was used (via git commit identifier), information regarding the duration of the run, and when it was started. Moreover, information regarding the computing node, OS, and Python version is also available. Hyperparameters are also available for each run, as shown in Figure 10. This, along with the git commit and hardware specifications, eases reproducibility. Finally, W&B also automatically logs system metrics along the training process, as shown in Figure 11.

## D Benchmarking dataset

This section gives more details on our third contribution. It consists of a dataset of training results of algorithms in MORL-Baselines applied on various environments available in MO-Gymnasium.

---

[11]We also support generating weight vectors based on OLS and GPI-LS methods.

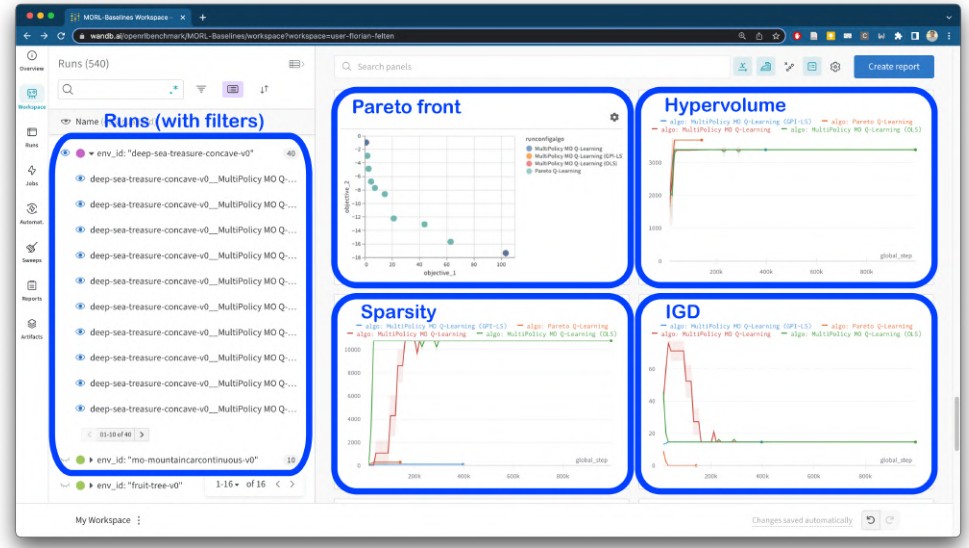

Figure 8: Screenshot of the W&B dashboards providing metrics and training information in real-time to the user.

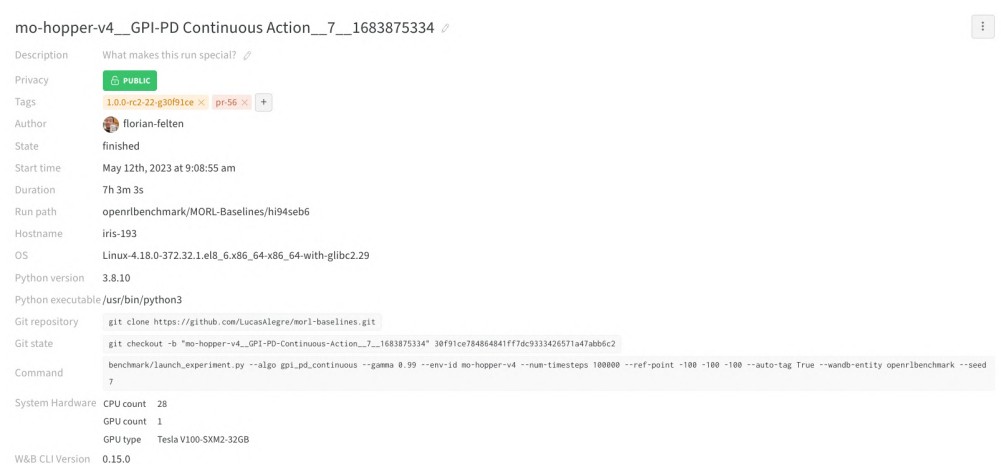

Figure 9: Overview of a given run in the W&B user interface. Notice that command line, OS and Python version, hardware specifications, and git commits are available for reproducibility.

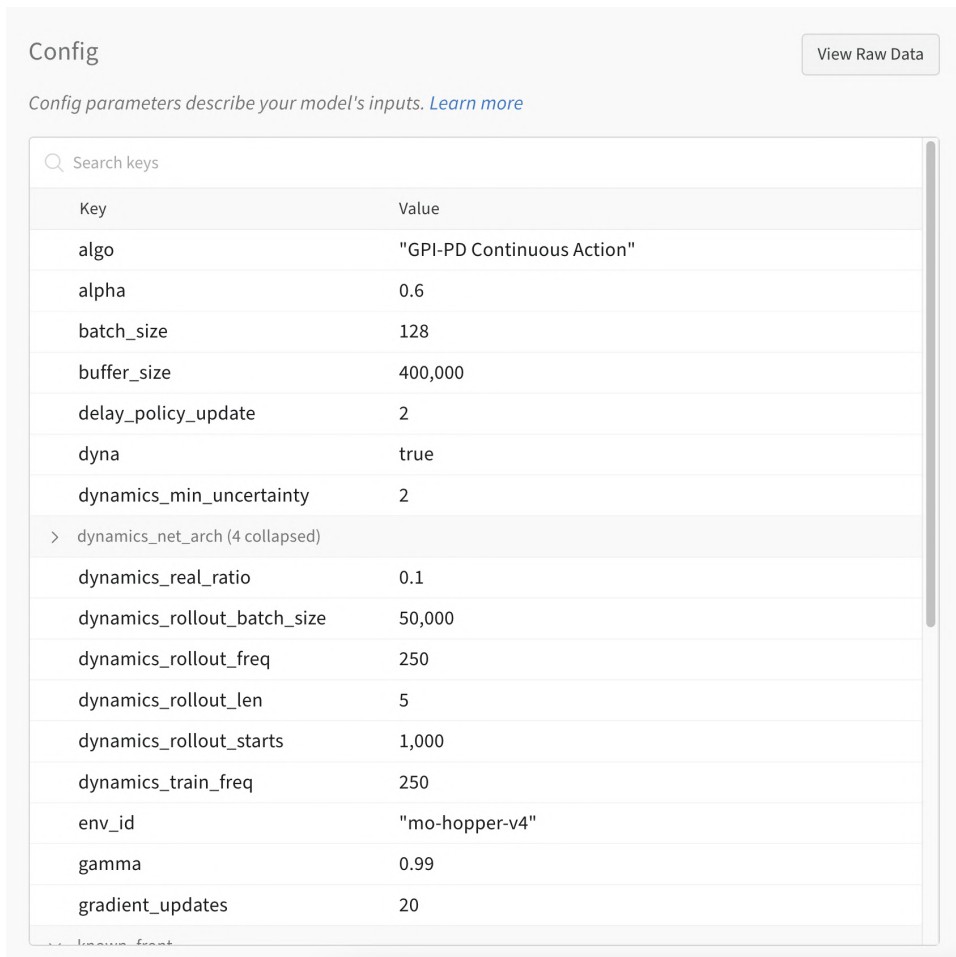

Figure 10: Hyperparameters of a given run.

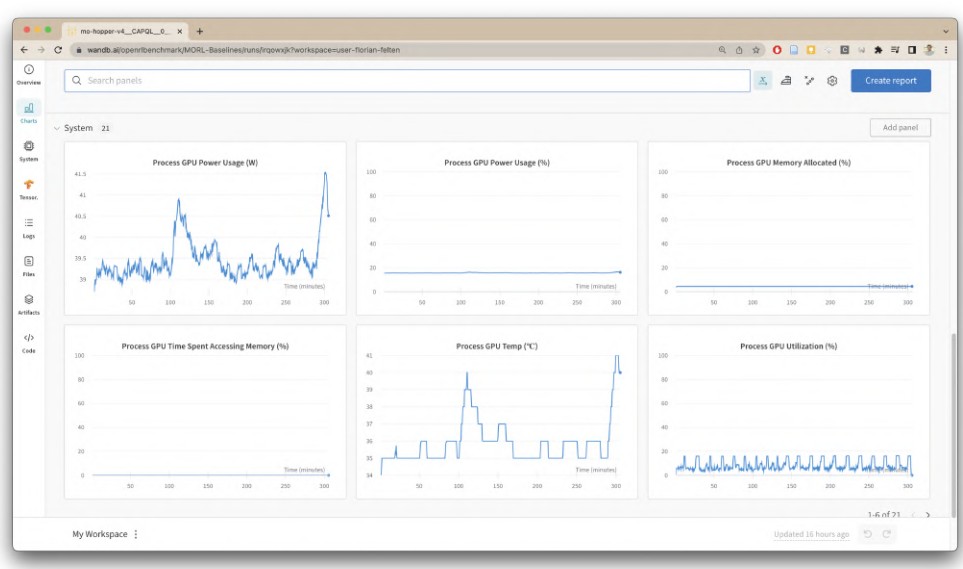

Figure 11: System usage metrics reported through W&B.

### D.1 Experimental settings

The section provides a more detailed description of the hardware and software settings used to produce our experimental results. For further details, the full specifications of each run can be found in W&B dashboards, as shown in Figure 9 and 10.

#### Hardware specifications

The experiments conducted on the University of Luxembourg high-performance computer (Varrette et al., 2014) were carried out on the *iris* cluster. The compute nodes equipped with GPU contain NVidia Tesla v100 SXM2 (16 and 32 GB) and Intel Xeon Gold 6132 @ 2.6GHz CPUs. Each training job has been allocated one GPU and 7 cores using SLURM (Yoo et al., 2003). The experiments conducted on the Vlaams Supercomputer Centrum (VSC) cluster Hydra hosted by the Vrije Universiteit Brussel (VUB) used nodes equipped with NVIDIA A100-PCIE-40GB GPUs.

Training all algorithms across all environments with multiple random seeds required approximately three months of computational resources.

#### Hyperparameters and code

All experiments have been run on MO-Gymnasium v1.0.0 and MORL-Baselines v1.0.0. All algorithms have been run with seeds from 0 to 9 on each supported environment. For most runs, we kept the default hyperparameters which are based on the original papers. However, in certain instances where we could not achieve satisfactory results, we made a minor attempt to fine-tune the hyperparameters. As a result, these hyperparameters, such as the Envelope hyperparameters, differ from those specified in the original paper. Finally, for the deterministic environments, we chose to increase the learning rate and exploration as these are the most critical in such domains. A GitHub issue (`https://github.com/LucasAlegre/morl-baselines/issues/43`) gives an overview of the command line used to launch each run, along with the values of the hyperparameters.

### D.2 Results

The following section presents the experimental results available in our dataset utilizing the command-line interface of openrlbenchmark (Huang et al., 2023). The provided command lines explicitly demonstrate how effortless it can be to generate plots to visualize the available evaluation metrics using this tool. It is important to note that openrlbenchmark is still in the developmental phase, and therefore its API is likely to undergo changes. Furthermore, we remind the reader that these results should be interpreted as proof of concept and should not be considered empirical evidence of performance differences among the implemented algorithms. In other words, we have not conducted enough runs to make statistically valid claims and have not fine-tuned the hyperparameters for all instances. Lastly, we would like to highlight that even though we have implemented single-policy algorithms, we have not extensively tested them at this stage.

#### Continuous observation and continuous actions

Figure 12a and 12b show the training results for environments with continuous actions and two objectives[12]. Both graphs show that PGMORL (Xu et al., 2020) seems less efficient than GPI-based algorithms (Alegre et al., 2023) or CAPQL (Lu et al., 2023) in terms of EU and Hypervolume. While CAPQL seems less sample efficient than GPI-based algorithms, the time-based plots (Figure 12b) look less categorical, especially for *mo-halfcheetah*. The sparsity metric is improved for CAPQL and PGMORL, as discussed in the main paper.

As a side note, we remind the reader that PGMORL is based on PPO (Schulman et al., 2017b) while the other algorithms are based on more modern single-objective RL algorithms. We believe that this plays a huge role in the final performances of the algorithms, *e.g.* if we were to reimplement PGMORL based on SAC (Haarnoja et al., 2018) or TD3 (Fujimoto et al., 2018b) we would probably obtain better performance. To make it easier to understand, we chose to keep

---

[12]Our implementation of PGMORL only supports environments with two objectives. This is because the implementation by the original authors relies on different methods for two and three objectives. Hence, PGMORL is only displayed in these two environments.

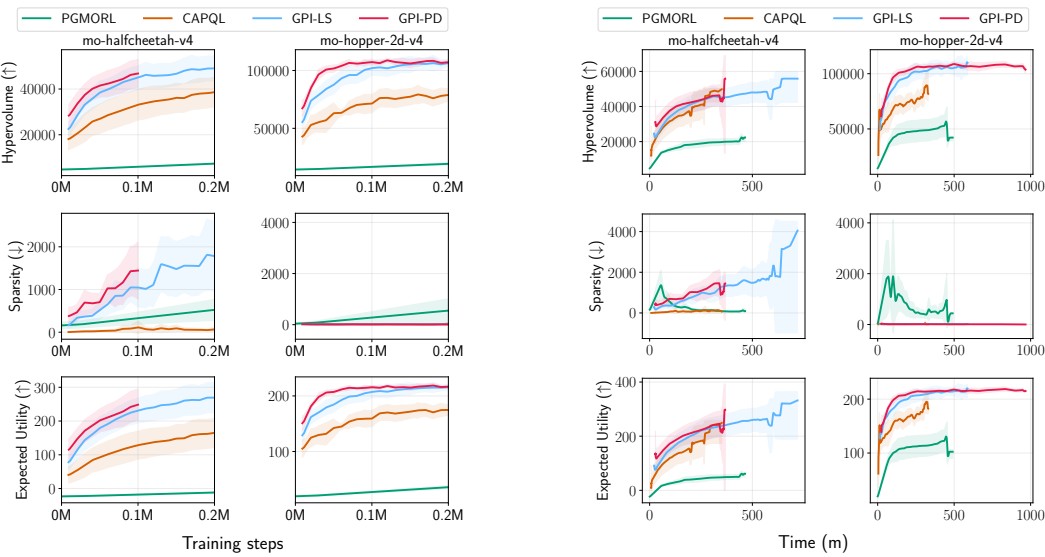

(a) Evaluation metrics w.r.t. training samples.

(b) Evaluation metrics w.r.t. training time.

Figure 12: Evaluation metrics in environments with continuous actions spaces.

faithful to the original papers instead. The following command was used to generate the plots:

```
python -m openrlbenchmark.rlops_multi_metrics \
    --filters '?we=openrlbenchmark&wpn=MORL-Baselines&ceik=env_id&cen=algo&metrics=
        eval/hypervolume&metrics=eval/sparsity&metrics=eval/eum' \
    'PGMORL?cl=PGMORL' \
    'CAPQL?cl=CAPQL' \
    'GPI-LS Continuous Action?cl=GPI-LS' \
    'GPI-PD Continuous Action?cl=GPI-PD' \
    --env-ids mo-halfcheetah-v4 mo-hopper-2d-v4 \
    --pc.ncols 2 \
    --pc.ncols-legend 4 \
    --pc.xlabel 'Training steps' \
    --pc.ylabel '' \
    --pc.max_steps 1000000 \
    --output-filename morl/morl_continuous \
    --scan-history
```

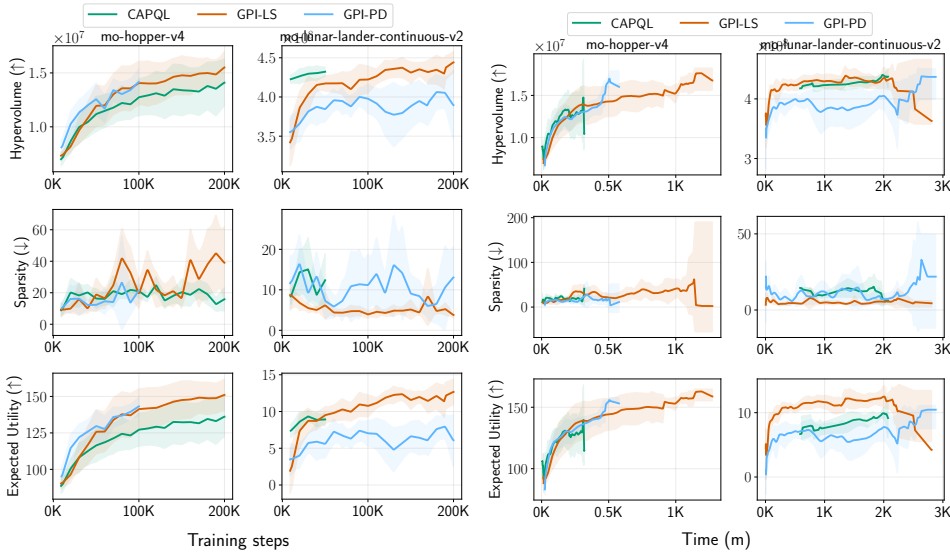

(a) Evaluation metrics w.r.t. training samples.    (b) Evaluation metrics w.r.t. training time.

Figure 13: Results on environments with more than two objectives and continuous observation and actions spaces.

The plots in Figure 13a and 13b give the training results for continuous environments with more than 2 objectives. In these environments, there are no clear-cut performance differences between the three algorithms. The plots were generated using the following command:

```
python -m openrlbenchmark.rlops_multi_metrics \
    --filters '?we=openrlbenchmark&wpn=MORL-Baselines&ceik=env_id&cen=algo&metrics=
        eval/hypervolume&metrics=eval/sparsity&metrics=eval/eum' \
    'CAPQL?cl=CAPQL' \
    'GPI-LS Continuous Action?cl=GPI-LS' \
    'GPI-PD Continuous Action?cl=GPI-PD' \
    --env-ids mo-hopper-v4 mo-lunar-lander-continuous-v2 \
    --pc.ncols 2 \
    --pc.ncols-legend 3 \
    --pc.xlabel 'Training steps' \
    --pc.ylabel '' \
    --output-filename morl/morl_continuous_more \
    --scan-history
```

**Continuous observations and discrete actions**

Figure 14 and 15 give the training results of MORL algorithms compatible with continuous observations and discrete actions. We can see that GPI-based methods outperform Envelope (Yang et al., 2019) and PCN (Reymond et al., 2022) for EU and Hypervolume. Yet, there is no performance difference between the model-based algorithm (GPI-PD) and the model-free one (GPI-LS) in this environment. It is worth noting that the performances of Envelope could be enhanced by tuning hyperparameters or using modern deep RL tricks such as dropout layers and normalization, as done in Alegre et al. (2023). Indeed, we believe that this algorithm suffers from pathologies that it inherits from DQN (Mnih et al., 2015a), such as overestimation bias. Finally, remember that PCN is designed for deterministic environments and thus is not really fit for the tested domains. Still, it performs

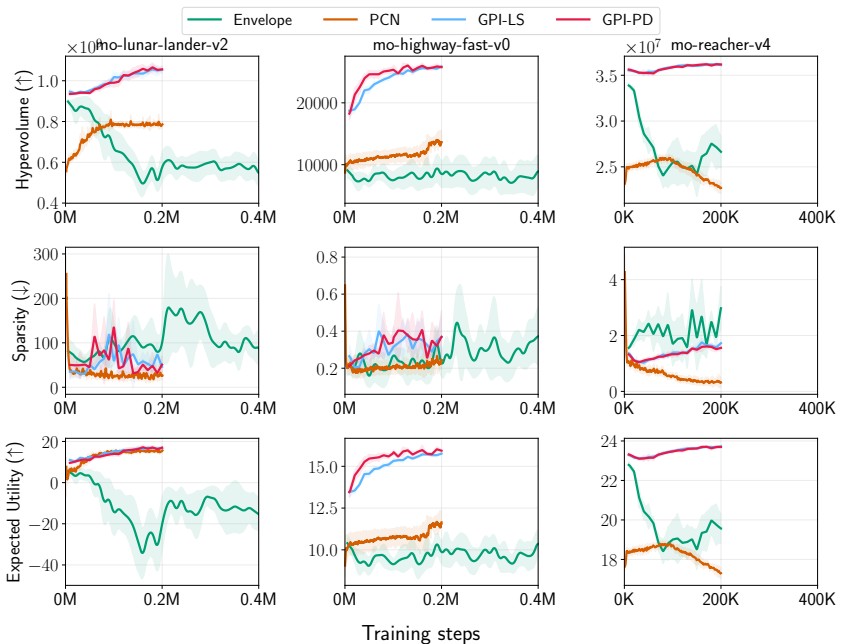

Figure 14: Evaluation metrics on environments with discrete actions and continuous observations w.r.t training samples.

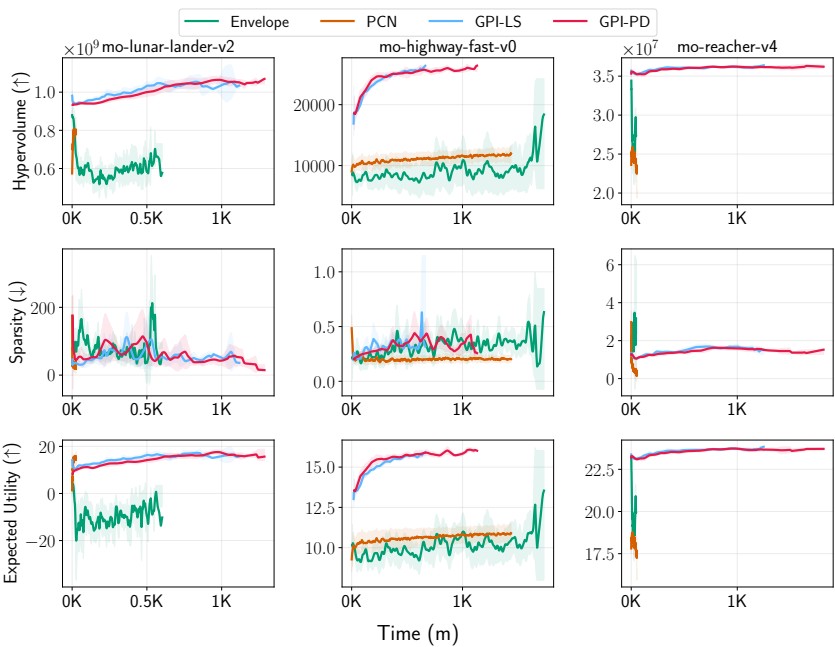

Figure 15: Evaluation metrics on environments with discrete actions and continuous observations w.r.t training time.

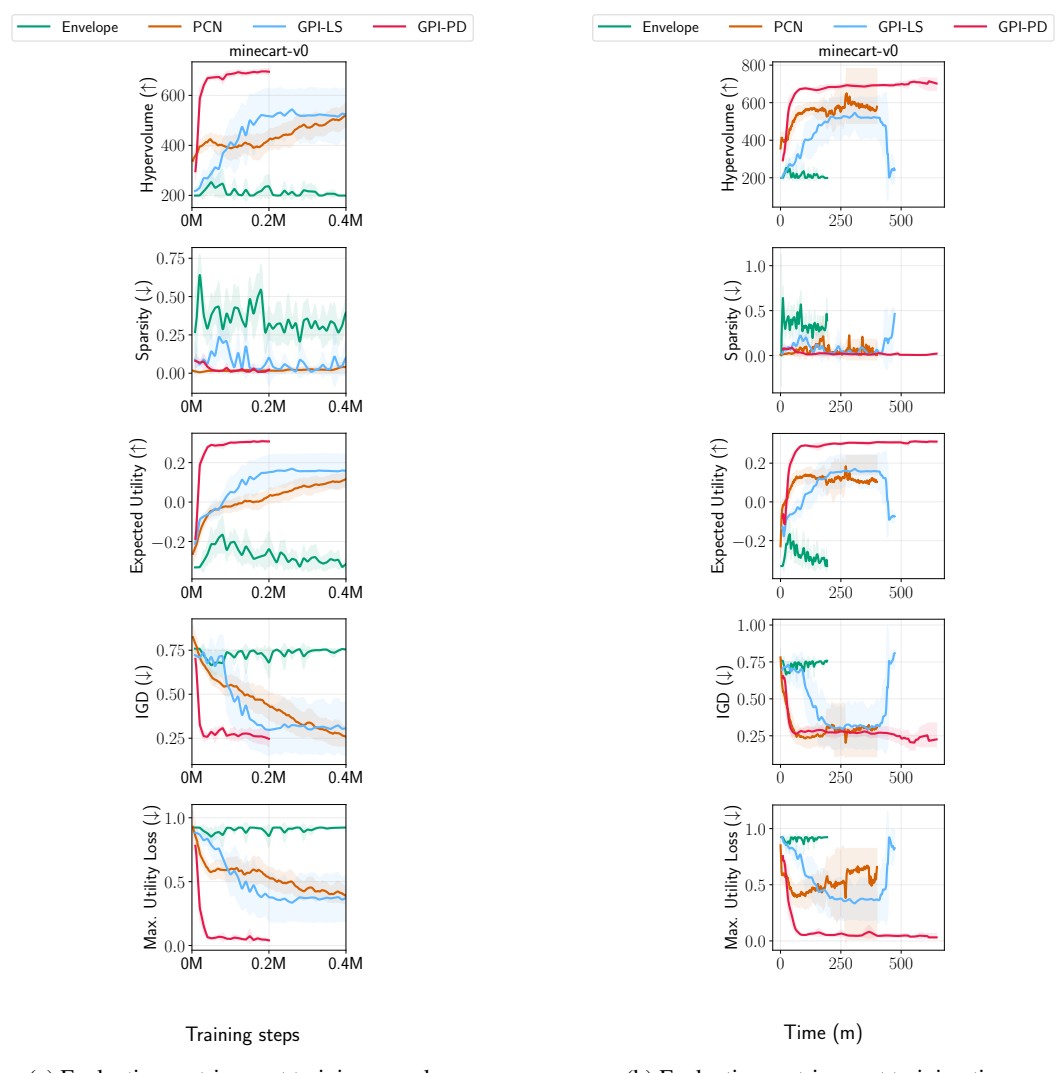

(a) Evaluation metrics w.r.t training samples.   (b) Evaluation metrics w.r.t training time.

Figure 16: Results on *minecart-v0*; discrete actions, continuous observations and known Pareto front.

surprisingly well on *mo-lunar-lander*. The plots were generated using the following command:

```
python -m openrlbenchmark.rlops_multi_metrics \
    --filters '?we=openrlbenchmark&wpn=MORL-Baselines&ceik=env_id&cen=algo&metrics=
        eval/hypervolume&metrics=eval/sparsity&metrics=eval/eum' \
    'Envelope?cl=Envelope' \
    'PCN?cl=PCN' \
    'GPI-LS?cl=GPI-LS' \
    'GPI-PD?cl=GPI-PD' \
    --env-ids mo-lunar-lander-v2 mo-highway-fast-v0 mo-reacher-v4 \
    --pc.ncols 3 \
    --pc.ncols-legend 4 \
    --pc.xlabel 'Training steps' \
    --pc.ylabel '' \
    --pc.max_steps 400000 \
    --output-filename morl/morl_discrete \
    --scan-history
```

The performance of the algorithms on the *minecart-v0* environment, which has a known Pareto front enabling the computation of IGD and MUL, is depicted in Figure 16a and Figure 16b. Similar

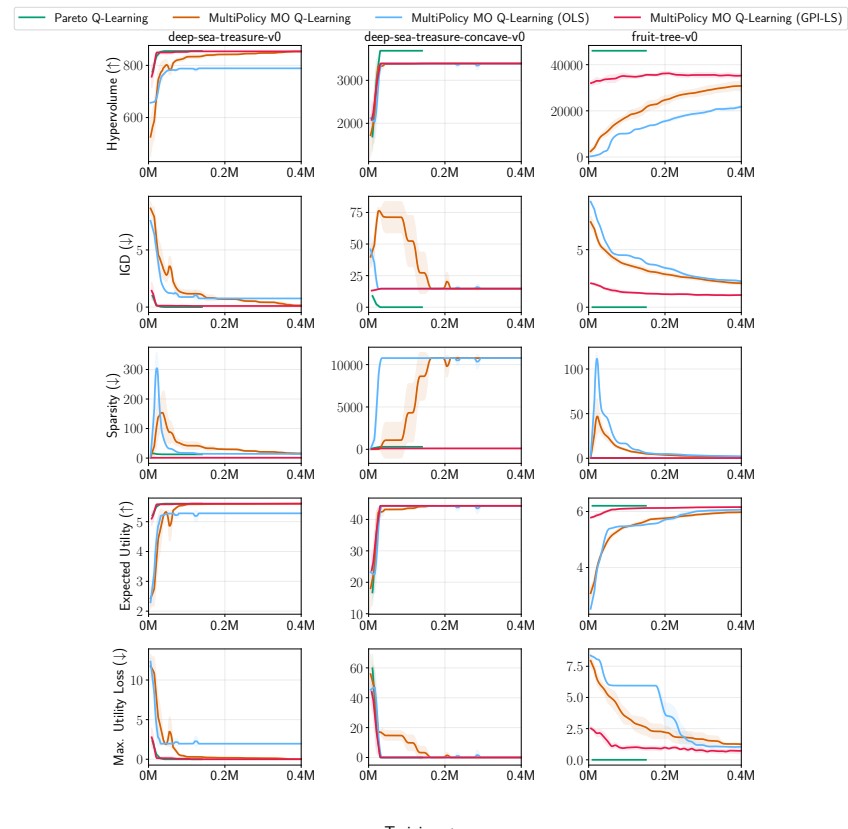

Figure 17: Evaluation metrics on deterministic environments with discrete actions and observations w.r.t. training samples.

to the previous plot, we can draw the same conclusion: GPI-based methods (especially GPI-PD) appear to be more sample efficient than Envelope and PCN. However, PCN demonstrates an advantage with its efficient implementation, making it indistinguishable from GPI-based methods when evaluating w.r.t. training time (Figure 16b). The following command line generated these plots:

```
python -m openrlbenchmark.rlops_multi_metrics \
    --filters '?we=openrlbenchmark&wpn=MORL-Baselines&ceik=env_id&cen=algo&metrics=
        eval/hypervolume&metrics=eval/sparsity&metrics=eval/eum&metrics=eval/igd&
        metrics=eval/mul' \
    'Envelope?cl=Envelope' \
    'PCN?cl=PCN' \
    'GPI-LS?cl=GPI-LS' \
    'GPI-PD?cl=GPI-PD' \
    --env-ids minecart-v0 \
    --pc.ncols 1 \
    --pc.ncols-legend 4 \
    --pc.xlabel 'Training steps' \
    --pc.ylabel '' \
    --pc.max_steps 400000 \
    --output-filename morl/morl_discrete_known_pf \
    --scan-history
```

**Discrete action and discrete observations**

Figure 17 and 18 illustrate the performance of tabular algorithms trained on environments with discrete actions and observations. Notice that these runs also include more metrics as they involve environments with known Pareto front. In all environments, Pareto Q-Learning (Van Moffaert and

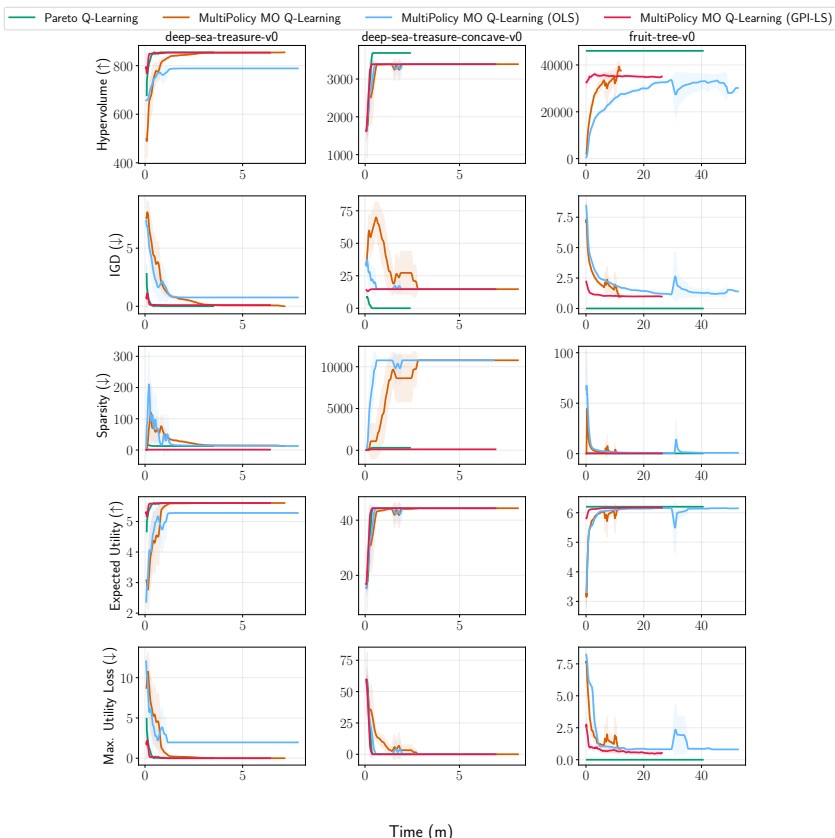

Figure 18: Evaluation metrics on deterministic environments with discrete actions and observations w.r.t. training time.

Nowé, 2014) finds optimal Pareto front, as shown by the IGD and MUL metrics reaching 0. Yet, our implementation of this algorithm is limited to deterministic environments; this could be enhanced by using the work of (Roijers et al., 2021). The MO Q-Learning (Van Moffaert et al., 2013a) variants, described in Section 5, seem to plateau in *deep-sea-treasure-concave* as they rely on linear scalarization and cannot find the points in the concave part of the front. Finally, the GPI-LS variant of MO Q-Learning shows strong performance in all environments compared to the other variants.

The plots were generated using the following command:

```
python -m openrlbenchmark.rlops_multi_metrics \
  --filters '?we=openrlbenchmark&wpn=MORL-Baselines&ceik=env_id&cen=algo&metrics=
      eval/hypervolume&metrics=eval/igd&metrics=eval/sparsity&metrics=eval/eum&
      metrics=eval/mul' \
  'Pareto Q-Learning?cl=Pareto Q-Learning' \
  'MultiPolicy MO Q-Learning?cl=MultiPolicy MO Q-Learning' \
  'MultiPolicy MO Q-Learning (OLS)?cl=MultiPolicy MO Q-Learning (OLS)' \
  'MultiPolicy MO Q-Learning (GPI-LS)?cl=MultiPolicy MO Q-Learning (GPI-LS)' \
  --env-ids deep-sea-treasure-v0 deep-sea-treasure-concave-v0 fruit-tree-v0 \
  --pc.ncols 3 \
  --pc.ncols-legend 4 \
  --pc.xlabel 'Training steps' \
  --pc.ylabel '' \
  --pc.max_steps 400000 \
  --output-filename morl/morl_deterministic_envs \
  --scan-history
```

